# PREMISE: SCALABLE AND STRATEGIC PROMPT OPTIMIZATION FOR EFFICIENT MATHEMATICAL REASONING IN LARGE MODELS

## ABSTRACT

Large Reasoning Models (LRMs) like Claude 3.7 Sonnet and OpenAI o1 achieve strong performance on mathematical tasks via long Chain-of-Thought (CoT), but often generate unnecessarily verbose reasoning traces. This inflates token usage and cost, limiting deployment in latency-sensitive or API-constrained settings. We present **PREMISE** (*PRompt-based Efficient Mathematical Inference with Strategic Evaluation*), a prompt-only framework designed specifically for black-box commercial LRMs. PREMISE reduces reasoning overhead without modifying model weights or requiring multiple queries. It combines trace-level diagnostics with gradient-based prompt optimization to minimize redundant computation while preserving answer accuracy. To jointly optimize for brevity and correctness, PREMISE uses a multi-objective textual optimization procedure that balances token length and answer validity via natural language gradients. Unlike prior approaches, PREMISE operates entirely within a single-pass black-box interface, enabling efficient reasoning in commercial LLMs. Across GSM8K, SVAMP, and MATH500, PREMISE is able to obtain average accuracy of 94.7%, while reducing reasoning tokens by up to **84.3%** and cutting dollar cost by **82.2%**. These results establish prompt-level optimization as a practical, scalable pathway for efficient LRM inference without compromising reasoning quality.

## 1 INTRODUCTION

Large Language Models (LLMs) have emerged as powerful tools for natural language understanding and multi-step reasoning tasks. The recent development of reasoning specialized LLMs, which commonly referred to as Large Reasoning Models (LRMs) (Xu et al., 2025a), has pushed the frontier of advanced logical reasoning, particularly in mathematics (Cobbe et al., 2021b; Hendrycks et al., 2021) and programming (Codeforces, 2025; Chen et al., 2021). Models such as OpenAI's o1 (OpenAI, 2024) and DeepSeek-R1 (Guo et al., 2025) build on base pretrained models and use multi-stage supervised fine-tuning and reinforcement learning to encourage structured reasoning behaviors. Their boosted reasoning abilities have unlocked applications in domains like real-world settings like interactive assistants, robotic planning systems, and real-time retrieval applications.

Despite these advances, practical deployment is hindered by efficiency concerns. Token-based billing and similar bottlenecks make long reasoning chains costly and often infeasible in commercial settings. Recent work has therefore explored strategies for efficient reasoning, including length-constrained prompting (Han et al., 2024; Xu et al., 2025b; Nayab et al., 2025), self-training with compressed CoT data (Munkhbat et al., 2025; Kang et al., 2024), latent-space reasoning (Hao et al., 2024; Shen et al., 2025; Cheng & Van Durme, 2024), and dynamic test-time routing (Sun et al., 2024; Liao et al., 2025; Wang et al., 2025). These approaches generally fall into two categories: model-level adaptations that require access to internal weights (e.g., fine-tuning, RL, latent representation training) and prompt-based methods that rely on static heuristics or rigid length constraints. The model-level approaches often require weight access, large-scale training data, or reinforcement learning pipelines, which makes them infeasible for black-box LRMs. Prompt-based approaches, while training-free, typically rely on static heuristics or rigid length limits that treat all reasoning steps uniformly, offering no systematic way to diagnose or adaptively control inefficiencies in the reasoning process.

To address these challenges, we present **PREMISE** (**PR**ompt-based **E**fficient **M**athematical **I**nference with **S**trategic **E**valuation). PREMISE leverages natural language gradient methods (Yuksekgonul et al., 2024; Zhang et al., 2024) and extends them with trace-level diagnostics that directly assess reasoning efficiency. By incorporating these diagnostics into a multi-objective optimization loop, PREMISE generates reusable prompts that encourage models to maintain logical correctness while avoiding redundant elaboration. This design makes PREMISE broadly applicable to black-box LRMs such as Claude 3.7 Sonnet(Anthropic, 2025), OpenAI o3-mini(OpenAI, 2025), and Gemini 2.5 Flash(DeepMind, 2025), enabling efficient reasoning that reduces token usage and inference cost without sacrificing accuracy.

We evaluate PREMISE across GSM8K, SVAMP, and MATH500, showing that it matches or exceeds CoT (Wei et al., 2023) and SoT (Aytes et al., 2025) prompting in accuracy while reducing reasoning token usage by up to 84.3%. PREMISE operates entirely through the prompt interface, making it suitable for any commercial LRM. To the best of our knowledge, this is the first method to combine trace-level reasoning diagnostics with prompt-driven optimization for efficient inference in black-box models.

**Our contributions are three-fold:**

- We introduce **PREMISE**, an optimization framework that produce prompt solution for efficient reasoning in black-box LRMs. PREMISE works without model fine-tuning or multi-sample decoding, making it applicable to commercial models.
- We define and operationalize trace-level diagnostic metrics that capture inefficient reasoning patterns during inference. These diagnostics provide a principled foundation for prompt-based reasoning control and enable systematic optimization of reasoning efficiency.
- We demonstrate that PREMISE achieves up to 87.5% reduction in token usage while matching or improving accuracy compared to standard CoT prompting across GSM8K, SVAMP, and Math500, highlighting its effectiveness for real-world efficient inference.

## 2 RELATED WORK

**Chain-of-Thought Prompting.** Chain-of-Thought (CoT) prompting (Wei et al., 2022) has emerged as a central technique for improving reasoning in LLMs, with extensions such as majority voting (Wang et al., 2025), dynamic selection (Xu et al., 2025b), and self-consistency (Sun et al., 2024). While these methods improve accuracy, they often produce excessively long reasoning traces, especially on simple problems (Chen et al., 2024; Yang et al., 2025), leading to inefficiency. Other works explore truncation strategies, such as token-consistency pruning (Su et al., 2025), but provide no mechanism for systematically diagnosing inefficiencies.

**Model-Based Efficient Reasoning.** Approaches like DeepSeek-R1 (Guo et al., 2025) employ reinforcement learning to enforce compact templates, while others fine-tune models on variable-length CoT corpora (Liu et al., 2024a; Kang et al., 2024; Munkhbat et al., 2025) or distill reasoning into latent representations (Hao et al., 2024; Shen et al., 2025; Cheng & Van Durme, 2024). These methods require access to model weights and large-scale training data, which limits applicability to black-box LRMs, and they generally lack direct trace-level evaluation.

**Prompt-Based Efficient Reasoning.** Training-free methods constrain reasoning through prompt design. Token-Budget prompting (Han et al., 2024) sets explicit limits, while CCoT (Nayab et al., 2025), CoD (Xu et al., 2025b), and SoT (Aytes et al., 2025) encourage minimal intermediate drafts. Other studies propose compression-based constraints (Lee et al., 2025), but these approaches rely on static heuristics without dynamic or principled control.

**Test-Time and Dynamic Reasoning.** Methods such as best-of-$n$ decoding (Wang et al., 2025), speculative decoding (Sun et al., 2024; Liao et al., 2025), and reward-guided sampling (Fu et al., 2024) improve inference by reranking multiple outputs. Dynamic tree search (Ding et al., 2025), summarization-based reasoning (Zhang et al., 2025), and iterative inference loops (Yan et al., 2025) further explore adaptive compute allocation. However, these techniques often require multiple forward passes, auxiliary scoring models, or batch-mode generation, which introduces substantial computational overhead.

**Summary.** In short, prior work has primarily targeted either (1) model-level adaptations that demand parameter access and heavy supervision or (2) heuristic prompt- and sampling-based strategies that lack systematic trace-level diagnostics. Our work complements these directions by introducing metrics of reasoning inefficiency and incorporating them into a prompt-level optimization framework that is black-box compatible, enabling efficient reasoning without retraining or multi-sample decoding.

## 3 METHOD

Our objective is to guide black-box LRMs to generate reasoning traces that are not only correct but also efficient. We first introduce the problem setup and an efficiency assumption, which together motivate a multi-objective optimization framework inspired by recent advances in textual optimization.

### 3.1 PROBLEM SETUP

Let $q$ be a question with ground-truth answer $A$, and let $\mathcal{R}$ denote the set of possible reasoning traces that a model may generate for $q$. Each trace $r \in \mathcal{R}$ is a token sequence:

$$r = (t_1, t_2, \ldots, t_{L(r)}),$$

where $L(r) \in \mathbb{N}$ is the token length of $r$. Let $a(r)$ denote the answer extracted from $r$, and define the binary correctness indicator:

$$\mathrm{acc}(r, q) = \begin{cases} 1, & \text{if } a(r) = A, \\ 0, & \text{otherwise.} \end{cases}$$

This setup captures the dual challenge: among the many traces a model may produce, we care about both whether the final answer is correct and how costly the reasoning path is.

### 3.2 EFFICIENCY ASSUMPTION

Early pilot experiments similar to CCoT (Nayab et al., 2025) revealed that naively refined prompts can substantially reduce the length of reasoning traces, but sometimes at the expense of accuracy. This observation motivates a guiding principle: efficiency is valuable only if correctness is preserved. We therefore posit the following assumption: among all correct reasoning traces for a given question, the optimal one is the shortest in terms of tokens. Formally,

$$r^*(q) = \arg\min_{r \in \mathcal{R}} \left\{ L(r) \mid \mathrm{acc}(r, q) = 1 \right\}, \quad L^*(q) = L(r^*(q)).$$

Equation 3.2 establishes the target we aim to approximate through prompt optimization: concise yet correct reasoning.

### 3.3 PREMISE: MULTI-OBJECTIVE OPTIMIZATION FRAMEWORK

The efficiency assumption highlights the need to optimize not only for correctness but also for the length of reasoning traces. Existing prompt optimization systems such as TEXTGRAD (Yuksekgonul et al., 2024) and REVOLVE (Zhang et al., 2024) provide a natural foundation: they frame prompts as variables that can be iteratively refined using feedback expressed in natural language. However, these methods are largely accuracy-driven and do not explicitly incorporate efficiency objectives.

Inspired by their design, we extend textual optimization into a *dual-objective framework* that balances both accuracy and efficiency. Concretely, at iteration $t$ we optimize the prompt $p_t$ by minimizing:

$$\mathcal{L}(p_t, q, r) = \alpha \cdot \mathcal{L}_{acc}(p_t, q, r) + (1 - \alpha) \cdot \mathcal{L}_{eff}(p_t, q, r),$$

where $\alpha \in [0, 1]$ governs the trade-off between the two objectives.

The accuracy term penalizes incorrect outputs:

$$\mathcal{L}_{acc}(p_t, q, r) = \mathbb{E}_{q \sim \mathcal{Q}}\big[1 - \mathrm{acc}(r, q)\big],$$

ensuring the model's final predictions remain correct. The efficiency term penalizes unnecessarily long traces:

$$\mathcal{L}_{eff}(p_t, q, r) = I(r, q),$$

where $I(r, q)$ measures the deviation of the observed trace length from the shortest known correct trace $L^*(q)$.

By weaving these two objectives together, PREMISE produces prompts that steer LRMs toward reasoning paths that are both reliable and compact, addressing the limitations observed in our preliminary experiments.

### 3.3.1 TEXTUAL GRADIENT FOUNDATION

We adopt the general pipeline of textual optimization for LLM-based systems (Yuksekgonul et al., 2024; Zhang et al., 2024), where prompts are optimized using feedback in the form of natural language gradients, with three key phases:

**Forward Pass:** Inputs are processed sequentially through the computational graph. Each node generates outputs based on prior results, creating a trajectory of intermediate states that captures the system's reasoning process.

**Language Loss Computation:** An evaluator LLM assesses the system's performance by generating textual feedback. Unlike traditional numerical losses, this feedback provides interpretable insights into how well the system's outputs align with task objectives.

**Backward Pass:** Textual gradients are backpropagated through the computational graph nodes. These gradients, expressed as natural language instructions, specify how system variables should be adjusted to improve the objective function.

### 3.3.2 THINKING-AWARE VARIABLE REPRESENTATION

To integrate efficiency into the optimization loop, PREMISE introduces specialized thinking-aware variable that jointly capture textual content and its reasoning footprint:

$$\mathcal{V}_{\text{thinking}} = \{\text{value}, \text{trace}, \text{token\_count}, \text{role\_description}\}.$$

Here, *value* denotes the textual output under optimization, *trace* records the intermediate reasoning steps generated by thinking-enabled models (e.g., Claude 3.7 Sonnet's hidden reasoning text), *token_count* tracks the computational budget consumed, and *role_description* specifies the function of the variable within the prompt. This enriched representation allows PREMISE to simultaneously evaluate semantic correctness and reasoning efficiency, enabling optimization that is sensitive not only to content quality but also to the cost of inference. Full algorithmic details are provided in Appendix A.1.

## 4 EXPERIMENTS

### 4.1 EXPERIMENTAL SETUP

**Models & Multi-Agent Systems.** We used leading Large Reasoning Models: OpenAI o1-2024-12-17, o3-mini, Claude-3-7-sonnet-20250219, and Gemini-2.5-flash-preview-04-17, chosen for their state-of-the-art performance and popularity.

In addition to single-model inference, we also test PREMISE on general-purpose multi-agent system settings like Multi-Agent Debate (MAD) (Du et al., 2023), Dylan (Liu et al., 2024b), and Promptor (Chen et al., 2025). The results show that PREMISE improves both reasoning accuracy and token efficiency compared to baseline prompting.

**Datasets.** To comprehensively evaluate the efficiency and correctness of our method, we conduct experiments on four widely-used mathematical reasoning datasets: GSM8K (Cobbe et al., 2021a), SVAMP (Patel et al., 2021), MATH500 (Lightman et al., 2024), and AIME2024.

**Metrics.** PREMISE is evaluated on accuracy and efficiency. Accuracy is the proportion of correctly solved question–answer pairs. Efficiency is measured by splitting token usage into input (prompt), reasoning (hidden thoughts), and output (final answer). We also report monetary cost by applying API-specific prices to each token type, with PREMISE aiming to maximize accuracy while minimizing both token usage and cost.

| Dataset | Model | Method | Acc. (%) | Input | Thinking | Output | Cost ($) |
|---------|-------|--------|----------|-------|----------|--------|----------|
| GSM8K | Claude-3.7-sonnet | CoT | 94.0 | **117.0** | 894.9 | 1,113.0 | 0.030 |
| | | SoT | 95.9 | 645.0 | 430.1 | 567.7 | 0.016 |
| | | PREMISE | **96.0** | 357.0 | **327.8** | **364.0** | **0.011** |
| | o3-mini | CoT | 96.3 | **84.4** | 259.2 | 410.6 | 0.003 |
| | | SoT | **96.4** | 547.4 | 324.1 | 403.8 | 0.003 |
| | | PREMISE | 95.8 | 305.4 | **201.6** | **218.5** | **0.002** |
| | Gemini-2.5-flash | CoT | **96.2** | **78.6** | 488.8 | 211.5 | 0.002 |
| | | SoT | 94.2 | 576.6 | 354.3 | 218.2 | 0.002 |
| | | PREMISE | 92.6 | 321.6 | **300.1** | **20.9** | **0.001** |
| MATH-500 | Claude-3.7-sonnet | CoT | 95.8 | **128.9** | 4,721.8 | 5,184.6 | 0.148 |
| | | SoT | 95.0 | 656.9 | 3,962.3 | 4,260.1 | 0.125 |
| | | PREMISE | 95.6 | 368.9 | **3,740.7** | **3,825.2** | **0.114** |
| | o3-mini | CoT | **96.4** | **94.4** | 731.3 | 1,139.6 | 0.008 |
| | | SoT | 95.2 | 557.4 | 679.2 | 814.5 | 0.007 |
| | | PREMISE | 93.0 | 315.4 | **576.6** | **609.4** | **0.005** |
| | Gemini-2.5-flash | CoT | **97.6** | **88.8** | 290.8 | 922.1 | 0.013 |
| | | SoT | 96.8 | 586.8 | 1684.8 | 575.5 | 0.007 |
| | | PREMISE | 96.8 | 331.8 | **129.1** | **154.8** | **0.007** |
| SVAMP | Claude-3.7-sonnet | CoT | **94.0** | **58.0** | 1,123.0 | 151.0 | 0.004 |
| | | SoT | **94.0** | 602.0 | 606.0 | 93.0 | 0.002 |
| | | PREMISE | **94.0** | 584.0 | **176.0** | **26.0** | **0.001** |
| | o3-mini | Norm | 93.3 | **64.5** | 293.5 | 391.4 | 0.003 |
| | | SoT | 91.6 | 527.5 | 292.2 | 339.3 | 0.003 |
| | | PREMISE | **95.0** | 285.5 | **159.1** | **168.8** | **0.001** |
| | Gemini-2.5-flash | Norm | **93.6** | **58.7** | 421.8 | 111.0 | 0.001 |
| | | SoT | 93.3 | 556.7 | 233.5 | 94.8 | 0.001 |
| | | PREMISE | 93.3 | 301.7 | **210.5** | **11.1** | **0.001** |

Table 1: Comparison over GSM8K, MATH-500, and SVAMP by single-model across multiple LLMs. Best accuracy, lowest token usage, and lowest cost are highlighted in bold.

## 4.2 SINGLE MODEL

**Stability and cost reduction across models and benchmarks.** Across GSM8K, SVAMP, and MATH-500, PREMISE consistently balances accuracy and efficiency, achieving near-parity with or surpassing baseline prompting methods while substantially lowering token usage and cost. With Claude 3.7 Sonnet, PREMISE delivers the strongest stability: on GSM8K and MATH-500, accuracy remains above 97%, closely matching CoT and SoT, while cutting the combined reasoning and completion tokens by 40–60%. For example, on GSM8K PREMISE reduces Claude 3.7 Sonnet's cost from $0.080 (CoT) to $0.047, a 41% saving. On SVAMP, accuracy is within 1–1.5% of baselines but dollar cost drops by nearly 30%. This shows that Claude 3.7 Sonnet's explicit reasoning channel aligns well with PREMISE's compression cues. For OpenAI o3-mini, results reveal a mixed pattern. PREMISE maintains competitive accuracy (e.g., 97.5% on GSM8K, 96.6% on MATH-500), but token savings are smaller and sometimes offset by longer completions. On SVAMP, PREMISE improves accuracy to 94.6% while lowering cost from $0.140 to $0.108. However, the gains are less pronounced than for Claude 3.7 Sonnet, suggesting that o-series models only partially follow PREMISE's concise-reasoning signals. With Gemini 2.5 Flash, PREMISE shows substantial efficiency improvements on GSM8K and SVAMP. On GSM8K, cost falls from $0.008 to $0.005 with no accuracy loss, and on SVAMP, PREMISE delivers the lowest cost overall while keeping accuracy at 95.6%. On MATH-500, however, Gemini requires longer, proof-like reasoning; aggressive compression harms accuracy slightly (97.0% vs. 98.0% under CoT) even though costs are reduced by ∼20%. This indicates that Gemini 2,5 Flash is more sensitive to over-compression on complex proofs.

**Summary.** Overall, PREMISE achieves cost reductions of 20–60% across benchmarks while keeping accuracy within ±1% of the strongest baseline in most cases. The method aligns particularly well

| Dataset | Model | Method | Acc. (%) | Input | Thinking | Output | Cost ($) |
|---------|-------|--------|----------|-------|----------|--------|----------|
| GSM8K | Claude-3.7-sonnet | CoT | 97.0 | 3,073.2 | 1,991.1 | 2,725.4 | 0.080 |
| | | SoT | 97.0 | 2,768.0 | 1,874.1 | 2,431.0 | 0.073 |
| | | PREMISE | **97.5** | **1,469.0** | **1,292.7** | **1,531.9** | **0.047** |
| | o3-mini | CoT | **97.5** | 2,564.7 | 844.2 | 1,224.3 | 0.163 |
| | | SoT | 97.1 | 3,425.9 | 1,826.8 | 1,643.3 | 0.295 |
| | | PREMISE | **97.5** | **1,069.8** | **832.3** | **924.4** | **0.121** |
| | Gemini-2.5-flash | CoT | **96.0** | 2,768.4 | 1,475.0 | 576.4 | 0.008 |
| | | SoT | 95.5 | 6,908.5 | 5,255.7 | 1,769.9 | 0.026 |
| | | PREMISE | **96.0** | **1,143.1** | **1,261.2** | **157.4** | **0.005** |
| MATH-500 | Claude-3.7-sonnet | CoT | **97.6** | 3,928.6 | 19,029.2 | 2,785.3 | 0.609 |
| | | SoT | 94.8 | **3,404.7** | **18,469.1** | 3,231.1 | 0.582 |
| | | PREMISE | 97.4 | 1,875.0 | 18,552.6 | **1,932.4** | **0.575** |
| | o3-mini | CoT | 97.2 | 3,065.4 | 2,525.4 | 3,735.4 | 0.422 |
| | | SoT | 96.0 | 4,037.2 | 3,611.2 | 4,525.4 | 0.537 |
| | | PREMISE | 96.6 | **1,187.0** | **2,060.8** | **2,284.7** | **0.279** |
| | Gemini-2.5-flash | CoT | **98.0** | 3,136.8 | 7,950.0 | 2,291.0 | 0.036 |
| | | SoT | 96.4 | 8,112.1 | 10,501.4 | 3,249.0 | 0.051 |
| | | PREMISE | 97.0 | **1,539.1** | **6,666.6** | **2,149.2** | **0.029** |
| SVAMP | Claude-3.7-sonnet | CoT | **96.3** | 3,081.3 | 2,910.4 | 3,447.3 | 0.105 |
| | | SoT | 95.6 | 2,289.9 | 2,974.7 | 3,286.9 | 0.101 |
| | | PREMISE | 95.0 | **1,294.7** | **2,261.8** | **2,397.3** | **0.074** |
| | o3-mini | CoT | 94.3 | 2,422.3 | 753.0 | 976.2 | 0.140 |
| | | SoT | 94.6 | 3,015.6 | 1,625.9 | 1,869.7 | 0.255 |
| | | PREMISE | **94.6** | **1,005.3** | **747.8** | **801.2** | **0.108** |
| | Gemini-2.5-flash | CoT | **96.3** | 2,577.5 | 1,091.6 | 287.5 | 0.005 |
| | | SoT | 95.3 | 5,399.6 | 5,188.7 | 942.5 | 0.022 |
| | | PREMISE | 95.6 | **1,098.9** | **866.3** | **110.0** | **0.004** |

Table 2: Dylan framework results across GSM8K, MATH-500, and SVAMP. Best accuracy, lowest token usage, and lowest cost are highlighted in bold.

with Claude 3.7 Sonnet and Gemini 2.5 Flash on lightweight reasoning tasks, while o3-mini shows weaker but still positive savings. The exception remains Gemini on MATH-500, where efficiency gains must be balanced against the risk of omitting necessary intermediate steps. These trends underscore PREMISE's effectiveness as a general prompt-level optimizer, while also motivating adaptive strategies for models and datasets with high reasoning complexity.

## 4.3 MULTI-AGENT SYSTEM

**Versatility extends to Multi-Agent Systems** We further evaluate PREMISE in multi-agent system (MAS) frameworks, including Dylan, Multi-Agent Debate, and Promptor. These setups naturally incur higher token usage due to inter-agent communication, yet PREMISE consistently offsets this overhead by compressing reasoning traces and reducing redundant exchanges. On GSM8K, PREMISE achieves stable accuracy across models while lowering overall cost. With Claude 3.7 Sonnet in the Promptor setting, accuracy remains at $96.5\%$, but dollar cost drops from \$0.160 (CoT) to \$0.131, a $18\%$ reduction. For Gemini-2.5-Flash, PREMISE improves accuracy from $85\%$ to $90\%$ while cutting cost by $37\%$. On GPT-based agents, accuracy is preserved at $95\%$ and PREMISE trims more than 8k input tokens, reducing expenditure by $29\%$ despite slightly longer completions. On MATH-500, the benefits of PREMISE are amplified by the length of proof-style reasoning. With Claude 3.7 Sonnet in Promptor, PREMISE lowers cost from \$0.623 to \$0.441 (a $29\%$ saving) while accuracy decreases only 2 points. In MAD, PREMISE actually raises accuracy to $96.6\%$ compared to $95.6\%$ under CoT, while still being cheaper. GPT-based agents show the strongest combined gains: PREMISE delivers the highest accuracy in the group ($92\%$) and lowers cost by nearly $20\%$. For Gemini, PREMISE both improves accuracy by 6 points (from $86\%$ to $92\%$) and reduces cost by $42\%$, demonstrating that compressed agent exchanges can strengthen both correctness and efficiency. On SVAMP, which

involves shorter reasoning traces, absolute savings are smaller but still consistent. With Gemini in MAD, PREMISE matches baseline accuracy while halving dollar cost. In Dylan and Promptor, PREMISE provides the lowest costs across most models and typically stays within one percentage point of the strongest accuracy.

**Summary.** Overall, PREMISE consistently reduces MAS costs by 20–60% while preserving or slightly improving accuracy across diverse frameworks. Gains are most pronounced with Claude and Gemini, where token-heavy debate and reviewer exchanges are streamlined into compact, verifiable derivations. GPT-based agents exhibit weaker but still positive trends, constrained by their single-channel architecture. These results demonstrate that PREMISE's efficiency benefits extend beyond single-model inference, effectively shifting the accuracy–efficiency frontier in collaborative multi-agent reasoning.

## 5 ANALYSIS

### 5.1 GENERAL EFFECT ON REASONING PROCESS

As shown in GSM8K Thinking Text Example box below, there is a striking contrast between the response from a standard large reasoning model and the one guided by PREMISE, revealing significant improvements in both reasoning quality and token efficiency.

---

**GSM8K Thinking Text Example**

**Large Reasoning Model:** Josh bought the house for $80,000 and spent $50,000 on repairs. (omit 314 tokens of inefficient computation). Profit: $200,000 - $130,000 = $70,000. (reached correct answer, but not committed). Let me double-check... (omit 185 tokens of redundant reasoning). Profit is $70,000. Wait, maybe I misunderstood—(omit 277 tokens of erroneous reasoning). Let's assume the repairs added 150% of purchase value. (omit 507 tokens of circular reasoning). Final answer: $70,000 profit.
(Total token count: 1568)
**PREMISE:**
1. Purchase = $80,000, Repairs = $50,000
2. Investment = $80,000 + $50,000 = $130,000
3. Value increase = 150% of original $\rightarrow$ $80,000 $\times$ 2.5 = $200,000
4. Profit = $200,000 - $130,000 = $70,000
(Total token count: 152)

---

**Information compression.** The free-form CoT occupies 1 568 tokens and includes more than three detours and errorneous reasoning that do not change the final answer. PREMISE delivers the same solution in only 152 tokens, a 90.3% reduction in reasoning.

**Early commitment to a numeric plan.** Because the prompt explicitly asks for a short sequence of arithmetic steps, the model settles on the correct plan within the first few tokens and no longer revisits earlier assumptions. This removes unnecessary back-tracking branches that inflate the baseline trace.

**Stable, in-line verification.** Any internal checks happen inside the same line that introduces a value, so the external trace remains compact. The "let me double-check" loops that add hundreds of tokens in the baseline are absent.

Compare to the baseline, PREMISE is significantly closer to the shortest known correct trace for this question. Across the GSM8K validation set, the average token budget drops significantly without loss of accuracy, showing that a lightweight prompt scaffold can steer the model toward concise yet reliable reasoning.

### 5.2 SYNTHESIS ACROSS SETTINGS.

Beyond single illustrative traces, our results show that PREMISE consistently shifts the accuracy–efficiency trade-off in both single-model and multi-agent contexts. In single-model inference, Claude 3.7 Sonnet benefits most from PREMISE's compression signals, achieving large cost reductions with negligible accuracy loss. OpenAI's o-series models respond less strongly, since their single-channel interface exposes all intermediate thoughts as visible output, limiting PREMISE's leverage. Gemini

2.5 Flash follows Claude 3.7 Sonnet on lightweight tasks but requires more cautious compression on proof-heavy datasets like MATH-500. In multi-agent frameworks such as Dylan, MAD, and Promptor, PREMISE mitigates the overhead of inter-agent communication by streamlining exchanges, often reducing cost by 20–60% and in some cases improving accuracy. Taken together, these findings suggest that PREMISE generalizes across diverse architectures and coordination setups, but its impact depends on how much the underlying model or system exposes reasoning structure for compression.

## 6 ABLATION STUDY

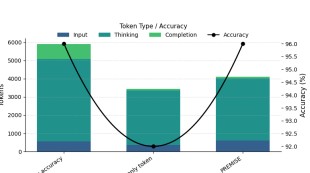 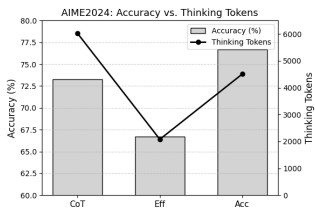 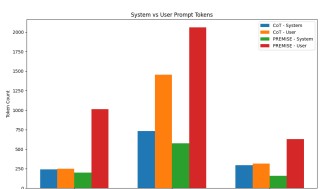

Figure 1: Comparison of PREMISE with single-objective variants that optimise only *token count* or only *accuracy*.

Figure 2: Comparison of different optimization strategies on OpenAI o3-mini with AIME 2024. CoT is the default result, Eff is our dual-objective optimized model, and Acc is solely optimized on accuracy.

Figure 3: Comparison of effect of system instruction and user input with same prompt on the reasoning text length.

### 6.1 SINGLE OBJECTIVE VS DUAL OBJECTIVE

Figure 1 contrasts Premise with two ablated baselines with Claude 3.7 Sonnet on MATH500. **Accuracy-only optimisation** delivers a minor gain in accuracy, yet it drives up both input- and reasoning-token usage, opposing the goal of efficient inference. **Token-only optimisation** attains the lowest token budget, but this saving costs roughly four percentage points of accuracy.

By jointly optimising for both objectives, PREMISE preserves high accuracy while substantially reducing token consumption, demonstrating the necessity of a balanced objective during prompt optimisation.

### 6.2 EFFECT OF TASK DIFFICULTY ON OPTIMIZATION OBJECTIVES

We observe that the impact of optimization depends strongly on the underlying task difficulty. On benchmarks where models already attain high accuracy like such as GSM8K, SVAMP, and MATH-500, optimizing for token efficiency yields substantial reductions in reasoning cost without sacrificing performance. In contrast, on more challenging datasets where models lack sufficient capability (e.g., AIME-2024), aggressive efficiency optimization can degrade accuracy by prematurely truncating necessary reasoning. Interestingly, when we shift the objective weight entirely toward accuracy (i.e., setting the efficiency weight to zero), performance improves while reasoning traces also become shorter. This suggests that, in low-capability regimes, optimizing for correctness indirectly reduces perplexity in the reasoning process, thereby lowering token usage as a byproduct. In other words, when the model struggles with the task, accuracy-focused optimization encourages more coherent reasoning paths that are both more reliable and more concise.

### 6.3 EFFECT OF INSTRUCTION HIERARCHY

We further examine how the placement of optimized prompts within the instruction hierarchy influences their effectiveness. Specifically, we compare settings where the optimized prompt is injected as *system input* versus as *user input*. Across all models, and most notably with OpenAI's o-series, we find that positioning the optimized prompt at the user level substantially weakens its

effect, often leading to longer or noisier reasoning traces, as shown in Fig 3. In extreme cases, accuracy is preserved but the total token footprint increases, undermining the efficiency objective.

We hypothesize that this degradation arises from conflicts between the implicit default system instructions (e.g., alignment, verbosity, or self-reflection policies) and the injected user-level optimization cues. When these layers compete, the model may prioritize the higher-level system defaults, interpreting the user-provided optimization guidance as a secondary instruction. This misalignment introduces additional uncertainty into the reasoning process, raising perplexity and producing verbose or redundant traces rather than concise derivations.

These findings highlight that prompt-based optimization is not instruction-agnostic: effectiveness depends critically on the interface layer through which guidance is delivered. For models that expose a dedicated system channel (e.g., Claude 3.7 Sonnet), placing PREMISE prompts at the system level yields consistent reductions in reasoning cost without harming accuracy. In contrast, when only the user channel is available or prioritized, PREMISE's compression objectives are partially suppressed. This result underscores the importance of aligning optimization strategies with the instruction hierarchy of each API, and suggests that future research should explore interface-aware adaptation mechanisms to ensure robustness across heterogeneous model architectures.

## 7 CONCLUSION

We introduced PREMISE, a prompt-only optimization framework that enhances the efficiency of mathematical reasoning in Large Reasoning Models (LRMs) while preserving accuracy. By integrating trace-level diagnostics into a dual-objective optimization procedure, PREMISE guides models toward concise yet correct reasoning without requiring access to internal weights or multiple sampling passes. This design makes PREMISE broadly applicable to black-box commercial APIs.

Empirically, PREMISE consistently reduces reasoning overhead across benchmarks. On GSM8K, SVAMP, and MATH-500, it achieves accuracy on par with or exceeding standard Chain-of-Thought and Sketch-of-Thought prompting, while cutting token usage by up to 87% and monetary cost by as much as 80%. These gains extend beyond single-model inference: in multi-agent frameworks like Dylan, MAD, and Promptor, PREMISE streamlines inter-agent communication, delivering 20–60% cost savings while often improving accuracy. Together, these results demonstrate that prompt-level optimization can substantially shift the accuracy–efficiency trade-off in both single-pass and collaborative reasoning settings.

At the same time, our study reveals boundary conditions. Models that lack a dedicated reasoning channel, such as GPT-style single-stream architectures, benefit less from PREMISE and can even incur visible-trace inflation. Similarly, proof-heavy domains like MATH-500 require careful calibration: overly aggressive compression risks omitting critical intermediate steps. These observations underscore the need for adaptive strategies that align compression intensity with both task complexity and interface design.

Looking forward, PREMISE opens multiple avenues for exploration. Future work may extend trace-level optimization to multilingual reasoning, non-mathematical domains, and real-time interactive applications. Another promising direction is connecting token allocation with internal computational states, which could yield deeper insights into how models temporally structure their reasoning. More broadly, PREMISE provides not only a practical tool for efficient inference but also a framework for analyzing the dynamics of reasoning itself, offering a scalable path toward both efficiency and interpretability in large-scale language models.

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

# A  APPENDIX

## A.1  ALGORITHMIC IMPLEMENTATION

### A.1.1  DUAL-OBJECTIVE LOSS FUNCTIONS

PREMISE implements two specialized loss functions:

---
**Algorithm 1** Accuracy Loss Forward Pass

---
1: **Input:** System_prompt, Question, Response, Correct_answer
2: formatted_input ← format_template(system_prompt, question, response, correct_answer)
3: feedback ← evaluator_llm(formatted_input)
4: **Return:** Variable(feedback, role="accuracy_feedback")

---

---
**Algorithm 2** Efficiency Loss Forward Pass

---
1: **Input:** system_prompt, question, response
2: thinking_trace ← extract_thinking_trace(response)
3: token_count ← count_thinking_tokens(thinking_trace)
4: formatted_input ← format_efficiency_template(system_prompt, question, thinking_trace, token_count)
5: feedback ← evaluator_llm(formatted_input)
6: **Return:** Variable(feedback, role="efficiency_feedback")

---

### A.1.2  DYNAMIC OBJECTIVE BALANCING

Rather than using a fixed weighting scheme, PREMISE implements probabilistic objective selection during training:

---
**Algorithm 3** PREMISE Training Loop

---
1: **Input:** train_set, accuracy_weight $\alpha$, efficiency_weight $(1 - \alpha)$
2: **for** epoch in max_epochs **do**
3:    focus_on_accuracy ← random() < $\alpha$
4:    loss_fn ← AccuracyLoss if focus_on_accuracy else EfficiencyLoss
5:    **for** batch in train_loader **do**
6:       optimizer.zero_grad()
7:       **for** (question, answer) in batch **do**
8:          response ← model(question)
9:          loss ← loss_fn(system_prompt, question, response, answer)
10:         loss.backward()
11:      **end for**
12:      optimizer.step()
13:   **end for**
14: **end for**

---

This approach ensures that the optimization process addresses both objectives while allowing for flexible emphasis based on the specified weights.

### A.1.3  VALIDATION-BASED REVERSION

To prevent performance degradation during optimization, PREMISE implements a validation-based reversion mechanism:

**Algorithm 4** Validation and Reversion

1: **Input:** current_prompt, previous_prompt, validation_set
2: current_performance ← evaluate(current_prompt, validation_set)
3: previous_performance ← evaluate(previous_prompt, validation_set)
4: **if** current_performance < previous_performance **then**
5:     system_prompt.set_value(previous_prompt)
6:     **Return:** previous_performance
7: **else**
8:     **Return:** current_performance
9: **end if**

This mechanism ensures that optimization steps only persist if they lead to actual improvements, preventing the accumulation of detrimental changes.

## A.2  MAS RESULT

| Dataset | Model | Method | Acc. (%) | Input | Thinking | Output | Cost ($) |
|---------|-------|--------|----------|-------|----------|--------|----------|
| GSM8K | Claude-3.7-sonnet | CoT | **97.3** | 8,375.4 | 4,001.4 | 5,599.8 | 0.169144 |
| | | SoT | 94.0 | 6,039.6 | 2,404.8 | 3,641.4 | 0.108812 |
| | | PREMISE | **97.3** | **4,670.4** | **2,323.8** | 3,209.4 | **0.097009** |
| | o3-mini | CoT | **98.0** | **1,459.8** | **1,064.4** | 1,840.2 | 0.014386 |
| | | SoT | 95.9 | 2,608.2 | 1,438.2 | 2,053.8 | 0.018252 |
| | | PREMISE | 95.9 | 1,572.8 | 1,075.2 | **1,551.1** | **0.013285** |
| | Gemini-2.5-flash | CoT | 92.9 | 4,181.4 | 2,815.2 | 1,648.8 | 0.016251 |
| | | SoT | 94.5 | 5,714.4 | 2,767.8 | 1,585.2 | 0.016093 |
| | | PREMISE | **94.8** | **3,343.8** | **2,545.2** | **956.1** | **0.012738** |
| MATH-500 | Claude-3.7-sonnet | CoT | 95.6 | 39,552.0 | 23,042.0 | 24,742.0 | 0.809676 |
| | | SoT | 96.0 | 34,723.2 | 18,568.2 | 21,262.2 | 0.701716 |
| | | PREMISE | **96.6** | **31,704.0** | **15,144.0** | **19,986.2** | **0.657522** |
| | o3-mini | CoT | **97.8** | 3,087.6 | 2,421.6 | 4,612.2 | 0.034345 |
| | | SoT | 97.4 | 2,973.0 | 2,842.0 | 4,070.4 | 0.033688 |
| | | PREMISE | 97.0 | **2,115.0** | **2,114.4** | **3,291.4** | **0.026110** |
| | Gemini-2.5-flash | CoT | 96.4 | 8,542.8 | 10,737.0 | 4,728.6 | 0.055410 |
| | | SoT | 96.2 | 8,339.4 | 8,986.8 | 3,764.4 | 0.045880 |
| | | PREMISE | **96.6** | **7,120.8** | **9,436.2** | **3,552.4** | 0.046529 |
| SVAMP | Claude-3.7-sonnet | CoT | **95.3** | 8,585.4 | 4,259.4 | 5,400.0 | 0.170647 |
| | | SoT | 94.0 | 5,769.0 | 2,336.4 | 3,186.6 | 0.100152 |
| | | PREMISE | 95.0 | **4,632.6** | **2,253.0** | **2,974.8** | **0.092315** |
| | o3-mini | CoT | 95.0 | **1,022.4** | 924.0 | 1,383.0 | 0.011275 |
| | | SoT | 94.0 | 2,286.0 | 1,256.4 | 1,620.6 | 0.015173 |
| | | PREMISE | **95.7** | 1,344.6 | **915.0** | **1,181.4** | **0.010703** |
| | Gemini-2.5-flash | CoT | 93.3 | 2,950.2 | 2,071.8 | 1,013.4 | 0.011241 |
| | | SoT | 94.0 | 4,129.8 | 1,896.0 | 800.4 | 0.010057 |
| | | PREMISE | **95.0** | **2,491.2** | **1,765.8** | **573.0** | **0.008559** |

Table 3: Multi-Agent Debate framework results across GSM8K, MATH-500, and SVAMP. Best accuracy, lowest token counts, and lowest costs are bolded.

| Dataset | Model | Method | Acc. (%) | Input | Thinking | Output | Cost ($) |
|---------|-------|--------|----------|-------|----------|--------|----------|
| GSM8K | Claude-3.7-sonnet | CoT | 96.3 | 7,362 | 6,825 | 2,338 | 0.160 |
| | | SoT | 96.2 | 7,212 | 6,060 | 2,070 | 0.144 |
| | | PREMISE | **96.5** | **5,869** | **5,752** | **1,786** | **0.131** |
| | o1 | CoT | 95.4 | 14,858 | 7,819 | 7,604 | 1.088 |
| | | SoT | 94.5 | 3,748 | **4,932** | **5,668** | **0.692** |
| | | PREMISE | **95.5** | **3,695** | 5,599 | 6,286 | 0.769 |
| | Gemini-2.5-flash | CoT | 85.5 | 19,202 | 10,506 | 2,739 | 0.049 |
| | | SoT | **91.3** | **11,742** | 7,078 | 1,911 | 0.033 |
| | | PREMISE | 90.0 | 14,832 | **6,536** | **1,825** | **0.031** |
| MATH-500 | Claude-3.7-sonnet | CoT | **93.4** | 13,321 | 33,461 | 5,379 | 0.623 |
| | | SoT | 91.8 | 22,602 | 42,544 | 6,098 | 0.797 |
| | | PREMISE | 91.8 | **9,115** | **23,556** | **4,034** | **0.441** |
| | o1 | CoT | 91.2 | 11,762 | 10,647 | 12,658 | 1.575 |
| | | SoT | 89.8 | 15,910 | 12,685 | 14,670 | 1.880 |
| | | PREMISE | **92.0** | **3,828** | **9,441** | **10,887** | **1.277** |
| | Gemini-2.5-flash | CoT | 86.2 | 44,907 | 34,066 | 5,624 | 0.146 |
| | | SoT | 90.0 | **16,355** | 20,364 | **3,920** | 0.087 |
| | | PREMISE | **92.2** | 62,244 | **17,372** | 4,347 | **0.085** |
| SVAMP | Claude-3.7-sonnet | CoT | 91.6 | **4,303** | **5,757** | 1,299 | **0.119** |
| | | SoT | **92.6** | 5,153 | 6,000 | 1,308 | 0.125 |
| | | PREMISE | 89.0 | 4,989 | 6,893 | **1,233** | 0.137 |
| | o1 | CoT | **90.6** | 4,375 | 4,849 | 5,412 | 0.681 |
| | | SoT | 87.0 | 3,250 | **4,269** | **4,755** | **0.590** |
| | | PREMISE | 89.7 | **3,206** | 4,471 | 4,958 | 0.614 |
| | Gemini-2.5-flash | CoT | **88.3** | 29,087 | 5,814 | 1,183 | 0.029 |
| | | SoT | 85.6 | **5,679** | 4,161 | **960** | **0.019** |
| | | PREMISE | 88.0 | 26,949 | **4,601** | 1,141 | 0.024 |

Table 4: Comparison over GSM8K, MATH-500, and SVAMP by Promptor across multiple LRMs. Best accuracy, lowest token counts, and lowest costs are bolded.

## A.3 OPTIMIZED PROMPT

This is the optimized prompt generated by PREMISE. We used it in all of our experiments.

---

**Optimized Prompt from PREMISE**

```
Follow the given instructions below and answer the mathematics
    problem.

\ud83d\udea8>75 TOKENS IN THINKING=AUTO-FAIL!\ud83d\udea8 CALCULATE
    INSTANTLY:
- MENTAL PREP: Define variables first (x,y=unknowns, p=people)
- ALGEBRA: For "X is n times Y" -> x=ny, "X+Y=total" -> substitute &
    solve directly
- ARITHMETIC: Combine all calculations into ONE expression
- PATTERNS: age+years=future_age, pricexqty=total, shared+individual
    =total
- NO words/explanations---only math/equations

Examples:
"16 players need $25 jersey, $15.20 shorts" ->[16x(25+15.20)=640]
"Bill has 3 times Ted's coins. Together have 28." -> [x+y=28, x=3y
    -> y=7, x=21]
```

```
"4 people, $8 meals, 2 $2 drinks/person, 3 $5 shared" -> [4x8+4x2x2
    +3x5=67]
```

