# OpenReview forum: "PREMISE: Scalable and Strategic Prompt Optimization for Efficient Mathematical Reasoning in Large Models"
_ICLR.cc/2026/Conference — Submitted to ICLR 2026_

### Official Review · Reviewer_w42D · 2025-10-19

**Soundness:** 3
**Presentation:** 3
**Contribution:** 3
**Rating:** 4
**Confidence:** 3

**Summary:**

This paper provides PREMISE, a prompt-only framework, for thinking token optimization while pertaining a comparable accuracy among different math benchmarks.

**Strengths:**

- Prompt-based, not necessarily to access the weight of the model
- Cut reasoning tokens down by 84.3% and cut dollar cost by 82.2% while having 94.7% accuracy.

**Weaknesses:**

## Major
- Line 160, $I(r, q)$ is not clearly defined, how can you calculate this metric? You've only mentioned this measures the deviation, is this term differentiable? How is it defined?
- Line 163, can you explain more about "shortest known correct" trace? How to know this metric without any prior knowledge?
- Can you report the result of not using any thinking token, like the standard IO prompting?
- Have you tried other benchmarks in Math or Logic? My impression of model performances on GSM8k and MATH-500 are pretty saturated.

## Minor
- Line 124, need to define $t_i$ properly.
- Line 154, use \textit{eff} than eff directly.

**Questions:**

- Line 15, "unnecessarily" seems debatable to me. Line 43 -- 44 need proper citations.

---

### Official Review · Reviewer_axik · 2025-10-28

**Soundness:** 2
**Presentation:** 2
**Contribution:** 2
**Rating:** 2
**Confidence:** 4

**Summary:**

This paper proposes a framework to opitmize LLM reasoning, and improve efficiency.

**Strengths:**

The framework is interesting, and the result shows promising performances.

**Weaknesses:**

This paper presents score improvement, however, it lacks technical details as well as baselines. I think it does not reach the high-level requirement of this conference.

1. I can not find the workflow or systematical presentation of their method. Therefore, it is hard for me to capture the technical details and contributions. The authors should include such figure in the manuscript. This is also not a theory paper I believe.

2. The baseline is not well-constructed. 1. The authors only consider small-scale LLM, and no open-source model. They need to include 5-6 models with different scales to support the conclusion is generalizable. The benchmarking setting is also not consistent. What is Norm in gemini-flash? Why it is not included in other dataserts?

3. The dataset used for mathematical reasoning is also outdated. The author should include more diverse mathematical datasets to perform reasoning benchmark.

4. Even if the authors do not consider the importance of new datasets and baselines, the improvement is not supersing, especially in the level of cost.

**Questions:**

Please see the weaknesses.

---

### Official Review · Reviewer_UB7Y · 2025-10-29

**Soundness:** 2
**Presentation:** 3
**Contribution:** 2
**Rating:** 2
**Confidence:** 4

**Summary:**

The paper introduces PREMISE, a prompt-only framework designed to improve the efficiency of mathematical reasoning in black-box Large Reasoning Models (LRMs). The challenge it addresses is that while CoT prompting achieves strong mathematical performance, it often generates verbose reasoning traces, leading to high token usage.

PREMISE uses a multi-objective textual optimization procedure that employs natural language gradients. This method refines the prompt to jointly optimize for two objectives: correctness (maintaining answer validity) and brevity (minimizing token length). The major contribution is a solution for efficient LRM inference that requires no modification of model weights or multiple queries, operating entirely within a single-pass black-box interface, making it suitable for commercial APIs. Across benchmarks like GSM8K, SVAMP, and MATH500, PREMISE achieves competitive accuracy while demonstrating substantial efficiency gains.

**Strengths:**

* The framework substantially reduces reasoning tokens by up to 84.3 percent, which translates to a reduction in monetary cost by as much as 82.2 percent. This makes Large Reasoning Models (LRMs) more viable for large-scale, cost-sensitive deployment.

* PREMISE is a prompt-only method that requires no modification of the model's weights. It is crucial for deployment with commercial, proprietary LRM APIs where model internal access is restricted.

* It employs a sophisticated multi-objective textual optimization procedure using natural language gradients. This method balances two goals: maintaining high correctness while aggressively improving brevity, a novel and effective approach to prompt engineering.

**Weaknesses:**

* The evaluation primarily focuses on relatively straightforward math datasets like GSM8K and SVAMP. The optimized prompt, which heavily emphasizes brevity and calculation-only output (A.3), may struggle with complex problems (like AIME or HMMT) that genuinely require and benefit from detailed, longer Chain-of-Thought reasoning.

* The core method is based on TextGrad, which has been previously established for prompt optimization. While the application is novel for efficiency, the paper's optimization mechanism is a relatively straightforward application of this existing technique to jointly target correctness and brevity.

**Questions:**

1. When calculating token costs, do you include the token consumptions during the prompt optimization phase, or only the final inference with the optimized prompt? Including optimization costs could affect the overall efficiency claims.

---

### Meta-Review · Area_Chair_hXrh · 2025-12-23

**Summary:**

This paper provides PREMISE, a prompt-only framework, to improve the computational efficiency of mathematical reasoning in black-box Large Reasoning Models (LRMs), aiming to reduce tokens used in reasoning. It refines the prompt to jointly optimize objectives: correctness and brevity. Empirical results on GSM8K, SVAMP, and MATH500 show competitive accuracy while achieving efficiency gains.

**Reviewer Concerns:**

Reviewers questioned the novelty of the work, as it is a straightforward combination of two existing work on a specific problem with no new techniques.

Another shared concern among reviewers is the evaluation. Used benchmarks are simple and outdated. The optimized prompts basically asks the model to output a single calculated result, thus achieving good results, which cannot generalize to complex math datasets. Some reviewers also raised concerns on its baselines being too simple.

Reviewers also mentioned that the paper is hard to follow with unclear definitions.

No rebuttal was provided.

**Reviewer Scores:**

I do not think reviewers would have changed their score if they had been able to participate fully in the discussion.

---

### Decision · Program_Chairs · 2026-01-26

Reject